# Explicit Analysis of Nonuniform Irradiation Swelling Pressure Exerting on Dispersion Fuel Matrix Based on the Equivalent Inclusion Method

**DOI:** 10.3390/ma15093231

**Published:** 2022-04-29

**Authors:** Yingxuan Dong, Junnan Lv, Hong Zuo, Qun Li

**Affiliations:** 1State Key Laboratory for Strength and Vibration of Mechanical Structures, School of Aerospace, Xi’an Jiaotong University, Xi’an 710049, China; daisybeast@stu.xjtu.edu.cn; 2Science and Technology on Reactor Fuel and Materials Laboratory, Nuclear Power Institute of China, Chengdu 610213, China; junnanlv@foxmail.com

**Keywords:** failure, the equivalent inclusion method, dispersion fuel meat, porous fuel particle, interaction layer, irradiation swelling

## Abstract

Under irradiation, dispersion nuclear fuel meat consists of a three-phase composite of fuel particles surrounded by an interaction layer dispersed within a metal matrix. Nonuniform swelling pressures are exerted on the matrix, generated by irradiation swelling of the fuel particles. As these are considerable, they can cause matrix failure, but they are difficult to calculate. In this paper, taking into account thermal expansion, nonuniform fission pores and the interaction layer, nonuniform irradiation swelling pressure has been formulated, based on the equivalent inclusion method. By means of doubly equivalent transformations, a porous fuel particle, surrounded by an interaction layer, which is under irradiation, can be simplified as a homogeneous particle with the eigenstrain. With the aid of Green’s function, nonuniform irradiation swelling pressure can be numerically analyzed. The simulation results of swelling pressures are in good agreement with numerical calculations. Furthermore, several simplified examples have been given to investigate the factors of influence and the impact mechanisms. Conclusions are drawn that nonuniform irradiation swelling pressure can be analyzed numerically and adopted to explore matrix failure. It is identified that the number and locations of fission pores inside a fuel particle are key factors for nonuniformity of swelling pressures. The volatility of swelling pressures is aggravated by burnup, while the average values of swelling pressures are intensely affected by temperature. This work provides a perspective to investigate the strength and integrity of dispersion fuel meat under high burnup.

## 1. Introduction

By dispersing ceramic fuel particles into the metal matrix, metal-based ceramic dispersion fuel meat has advantages of low core temperature, inherent high safety, high radiation resistance, deep burn-up, long service life, etc. [1,2,3,4] Under irradiation, the dispersion nuclear fuel meat consists of fuel particles surrounded by an interaction layer and are dispersed within the metal matrix, which is equivalent to the three-phase inclusion problem [5]. There are massive fission fragments and heat generated by the fission reaction of nuclear fuel [6]. Under high burnup, uneven fission pores appear inside ceramic fuel particles, due to gaseous fission fragments [7]. At the same instant, there exists an interaction layer surrounding the fuel particles, which is induced by solid fission fragments [8,9]. The swelling of fuel particles is mainly caused by gaseous fission fragments, namely, the effect of internal pressures of uneven fission pores [10]. Under thermal expansion of fuel particles, there are intense mechanical interactions [5] among the three components of fuel meat, i.e., fuel particles, interaction layer and metal matrix. Simultaneously, the expanding fission pores enhance the deformations of fuel particles. Therefore, subjected to temperature, burnup and fission pores, nonuniform swelling pressures are exerted on the matrix, dependent on thermal expansion and fission gases of the ceramic fuel. The failure mechanism of dispersion fuel meat is affected greatly by this nonuniform swelling of fuel particles. However, it is difficult, but important, to calculate the nonuniform swelling pressure for such a dispersed structure under a severe irradiation environment. Dispersion fuel meat is the same as an inclusion problem for three-phase materials, which can be analyzed by the equivalent inclusion method [11]. Accordingly, the need for formulation and analysis of nonuniform swelling pressures exerted on the dispersion fuel matrix is considerable in investigating matrix failure, evaluating strength and assessing the integrity of dispersion fuel meat.

The equivalent inclusion method, which was proposed and developed by J. D. Eshelby and T. Mura [12,13,14], has been extensively researched as an effective approach for inhomogeneity problems, such as the physical fields induced by inhomogeneities [12,15], the overall elastic moduli and coefficients of thermal expansion of composite materials [16], inclusion problems with non-uniform axisymmetric eigenstrain distribution [17], physical fields at crack tip enclosed by a homogeneous inclusion [18,19], size effect analysis in inhomogeneous materials [20], etc. It is recognized that the equivalent inclusion method has advantages for analyzing composites under multi-field coupling conditions. By equivalent transformation, an inhomogeneous inclusion can be replaced by a homogeneous inclusion, subject to the eigenstrain. The equivalent eigenstrain can be formulated by the principle of virtual work [21]. Then the physical fields for the circular homogeneous inclusion problems can be characterized with the aid of eigenstrain and Green’s function. After transformation, the equivalent homogeneous inclusion problems have explicit solutions, in terms of the definite integral. Herewith, the stress and strain fields of inclusion problems within dispersion fuel meat are numerically solvable. However, existing research has not taken account of irradiation conditions in applying the equivalent inclusion method. For dispersion fuel elements, which possess complex material structures and mechanical-thermal-irradiated coupling fields, the equivalent inclusion theory must be supplemented by introducing the fission-gas strain field and the thermal strain field.

In view of the foregoing advantages, the equivalent inclusion method and Green’s function can be effectively employed to analyze the nonuniformity of irradiation-induced swelling effect associated with thermal expansion and fission gases of ceramic fuel. Through nonuniform analysis of irradiation swelling, the strength, failure and integrity problems of dispersion fuel meat can be investigated. In order to validate the accuracy of numerical results, finite element analysis can be carried out for the representative volume element model of fuel meat with single particle. Taking into account the temperature, burnup and interaction layer, the irradiation effects can be reflected into the formulation of swelling pressure of a fuel particle exerting pressure on the matrix. Furthermore, to improve computability and convergence, the effects of porous fuel particles on their surrounding matrix can be replaced by nonuniform swelling pressures as internal loading conditions in simulation of full-sized dispersion fuel elements, which saves on experimental consumption and raises efficiency in optimizing fuel elements.

In this paper, the nonuniform irradiation swelling pressure exerted on a matrix is formulated under given irradiation conditions. In Section 2, the general formulations for the equivalent inclusion method are reviewed. In Section 3, taking into account three-phase composite and irradiation conditions, the nonuniform irradiation swelling pressure $P_{f}$, associated with temperature and fission gases of ceramic fuel particles, is formulated by doubly equivalent transformations of a fuel particle surrounded by an interaction layer. Several simplified examples are calculated in Section 4 to demonstrate the analytical process of the obtained nonuniform irradiation swelling pressure. By comparison with finite element simulations, the accuracy of the presented formalism was validated. Subsequently, several analytical results have been discussed about the influences of the numbers and locations of fission pores, temperature and burnup. The representative volume element model for a porous fuel particle is analyzed with our presented method. In Section 5, some conclusions are drawn. The presented formalism for the nonuniform swelling pressure exerted on a matrix of dispersion fuel meat is expected to provide a feasible method to evaluate the failure of dispersion fuel meat, due to the synthesis effect of temperature, burnup, material properties, etc.

## 2. The Equivalent Inclusion Method

As illustrated in Figure 1, the inhomogeneous inclusion problem can be transformed into a homogeneous inclusion problem with eigenstrain by equivalent transformation. The eigenstrain is defined as the general name for stress-free strains, thermal expansion strains, plastic strains, phase transformation strains, residual strains, inhomogeneous strains, etc. [13]. Consider an infinite matrix, having fourth-order elastic tensor C, subjected to a stress-free strain $\varepsilon^{\prime }$ distribution in an inclusion which has the fourth-order elastic tensor $C^{*}$. The inhomogeneous system is equivalent to another that the matrix is subjected to, a second-order eigenstrain tensor $\varepsilon^{\prime \prime }$ in a homogeneous inclusion with elastic tensor C. Therefore, the derivation of physical fields in inclusions and matrix is the process by means of which to determine eigenstrain.

For inhomogeneous inclusion, the total stress tensor $\sigma^{I}$ is the sum of applied stress $\sigma^{0}$ and disturbance stress $\sigma^{\prime }$ due to stress-free strain $\varepsilon^{\prime }$. That is
(1)$$\sigma^{I}=\sigma^{0}+\sigma^{\prime }$$

According to Hooke’s law, the Equation (1) can be rewritten as
(2)$$\sigma^{I}=C^{*}:(\varepsilon^{0}+\varepsilon^{\prime })$$

With respect to equivalent homogeneous inclusion with eigenstrain $\varepsilon^{\prime \prime }$, the total stress $\sigma^{I}$ can be expressed as
(3)$$\sigma^{I}=C^{*}:(\varepsilon^{0}+\varepsilon^{\prime }-\varepsilon^{\prime \prime })$$

Hence, on the basis of the equivalent transformation as depicted in Figure 1, there must be same stress states in an inclusion before and after the transformation, i.e.,
(4)$$C^{*}:(\varepsilon^{0}+\varepsilon^{\prime })=C:(\varepsilon^{0}+\varepsilon^{\prime }-\varepsilon^{\prime \prime })$$
where the disturbance strain $\varepsilon^{\prime }$ can be defined as $\varepsilon^{\prime }=S^{-1}:\varepsilon^{r}$. S denotes the Eshelby’s tensor, defined as a function corresponding with the Poisson rate and inclusion shape [22]. And $\varepsilon^{r}$ is the restraint strain of the matrix on the inclusion. It is noted that the disturbance strain can also be presented according to real conditions.

## 3. Formalism of Irradiation Swelling Pressure by Equivalent Inclusion Method

Under irradiation, the dispersion fuel meat is equivalent to the three-phase inclusion model with thermal strain, mechanical strain and acts of fission fragments. As shown in Figure 2a,b, within the dispersion fuel meat, the single particle representative of the volume element is composed of a porous fuel particle, interaction layer and metal matrix.

There are internal pressures *P_g_* in fission pores produced by gaseous fission fragments. Subjected to solid fission fragments, there exists an interaction layer with thickness of *d*_IL_ between a fuel particle and matrix. Let Ω⊂ℝ3 denote the inclusion domain Ω of a fuel particle embedded in an infinite matrix in the three-dimensional Euclidean space ℝ3. Γ is the interface between the metal matrix and the interaction layer. And let an infinitely extended matrix be subjected to an eigenstrain ε∗∗ distribution of a porous fuel particle surrounded by the interaction layer, as depicted in Figure 2c. The factors, u, ε and σ denote second-order tensors of displacement, strain and stress, respectively. Due to environmental pressure *P_m_*, there exists the applied strain field of matrix ε0. The physical fields of interaction layer are uIL, εIL and σIL, respectively. The elastic tensors of matrix, interaction layer and fuel are CM, CIL, and Cf, respectively. After equivalent transformation, a porous fuel particle could be transformed as a homogeneous inclusion embedded in the matrix with the interaction layer and fission pores, as presented in Figure 2c. Under irradiation, there exists nonuniform swelling pressure Pf exerting itself on the matrix, produced by uneven expansion of the fuel particle. In this section, according to the equivalent inclusion method, the composite of a porous fuel particle surrounded by an interaction layer was transformed into a homogeneous spherical inclusion with the eigenstrain ε∗∗, by doubly equivalent transformations, as shown in Figure 3. Then, the nonuniform irradiation swelling pressure Pf can be derived by the condition of strain continuity on the interface Γ between matrix and interaction layer.

Within a porous fuel particle, the quantities of fission pores ascend with increasing burnup. The diffusion coefficient of fission gas in the fuel phase is very small and hardly migrates from one fission pore to another through thermal activation diffusion [23]. Additionally, due to the inhomogeneous distribution of pores inside a fuel particle, the strain field of interaction layer is affected by the inhomogeneity strain field $\varepsilon^{H}$, fission gas strain field $\varepsilon^{G}$ and thermal strain field $\varepsilon^{T}$. Based on the equivalent inclusion method, a porous fuel particle can be transformed into a homogeneous particle subjected to the eigenstrain $\varepsilon^{*}$, as presented in Figure 3b. And there exists
(5)$$\varepsilon^{*}=\varepsilon^{H}+\varepsilon^{T}+\varepsilon^{G}+\varepsilon^{0}$$

Hence, through the first equivalent transformation, a porous fuel particle surrounded by an interaction layer has been translated into am homogeneous particle subjected to the eigenstrain $\varepsilon^{*}$ and surrounded by an interaction layer, as depicted in Figure 3b. The influence of uneven fission pores is eliminated. Moreover, the composite in Figure 3b can be transformed into a homogeneous particle with the second eigenstrain $\varepsilon^{**}$, as illustrated in Figure 3c. According to the equivalent inclusion method of Equation (4), an equilibrium condition exists, as follows
(6)$$C^{f}:(\varepsilon^{\mathrm{IL}}+\varepsilon^{*})=C^{\mathrm{IL}}:(\varepsilon^{\mathrm{IL}}+\varepsilon^{*}-\varepsilon^{**})$$

Substituting Equation (5) into (6) yields
(7)$$C^{f}:(\varepsilon^{\mathrm{IL}}+\varepsilon^{G}+\varepsilon^{H}+\varepsilon^{T}+\varepsilon^{0})=C^{\mathrm{IL}}:(\varepsilon^{\mathrm{IL}}+\varepsilon^{G}+\varepsilon^{H}+\varepsilon^{T}+\varepsilon^{0}-\varepsilon^{**})$$so, the eigenstrain $\varepsilon^{**}$ is derived as
(8)$$\varepsilon^{**}=[(C^{\mathrm{IL}}-C^{f}):{\left(C\mathrm{IL}\right)}^{-1}]:(\varepsilon^{\mathrm{IL}}+\varepsilon^{G}+\varepsilon^{H}+\varepsilon^{T}+\varepsilon^{0}).$$

After the second equivalent transformation, inhomogeneity between the fuel particles and interaction layer is eliminated.

It is known that the total displacement field of the interaction layer is equal to the displacement field induced by eigenstrain ε∗∗ of the homogeneous inclusion [14], that is
(9)u(x)=∫Ω∇⊗G(x−x′):(C:ε∗∗(x′)) dx′
where x′ and x denote the geometric positions of an influence point set with eigenstrain ε∗∗ and a research point set with induced strain ε in ℝ3, respectively. It submits to the relations of x′∈Ω and x∈ℝ3. ∇ is the gradient operator. And G(x) is Green’s function, which indicates a mathematical function for solving non-homogeneous differential equations with boundary or initial conditions [24]. It characterizes the interaction of source and field. Based on the Green’s functions of specific problems, amounts of practical problems can be easily represented and analyzed with integral equations [25,26,27].

According to the Voigt notation expression, the elastic tensor C can be presented as $C_{ijkl}$. For brevity, all tensors will follow the Voigt notation hereinafter.

On the isotropic assumption of the interaction layer, the relevant Green’s function is [28],
(10)Gij(x,x′)=14πμILδij|x−x′|−116πμIL(1−νIL)∂2∂xi∂xj|x−x′|in ℝ3
where δij is the Kronecker delta. νIL and μIL are Poisson’s ratio and the shear modulus, respectively. x(xi) and x′(xi) are functions of spatial coordinates xi. And it submits to |x−x′|2=(xi−xi′)(xi−xi′).

Combining Equations (8)–(10), the total strain field of interaction layer is expressed as
(11)$$\varepsilon (x)=\int_{\Omega }(C^{f}-C^{\mathrm{IL}}):(\varepsilon^{\mathrm{IL}}+\varepsilon^{G}+\varepsilon^{H}+\varepsilon^{T}+\varepsilon^{0}):\;\zeta (x,x^{\prime })\;dx^{\prime }$$
where ζ(x,x′) denotes a four-tensor influence function. In Voigt notation expression, there exists
(12)$$\zeta_{ijkl}(x,x\prime )=-\frac{1}{2}\left[G_{ik,lj}(x,x^{\prime })+G_{jk,li}(x,x^{\prime })\right]$$

The matrix deforms due to the act of eigenstrain of the fuel particles. There is irradiation swelling pressure $P_{f}$ exerted on the matrix. On the interface Γ in Figure 2c, the stress field is noncontinuous while the strain field is continuous, that is
(13)$$\varepsilon (x,x\in \Gamma )=\varepsilon^{M}(x,x\in \Gamma )$$
where $\varepsilon^{M}(x,x\in \Gamma )$ is the strain field on interface Γ of the matrix. The total strain field of the interaction layer is equal to the strain field of the metal matrix on the interface Γ. The stress field on the interface Γ of the matrix can be solved by
(14)$$\sigma^{M}(x,x\in \Gamma )=C^{M}\varepsilon^{M}(x,x\in \Gamma )$$

Ceramic materials generally possess elastic constitutive relations, even in irradiation environments [29]. The interaction layer between UO_2_ and metal materials possesses complex chemical compositions, which generally consist of oxides of metals and uranium [9]. Therefore, because the mechanical characterizations of the interaction layer are difficult to test and have not been reported, it will be considered as embrittlement metal after irradiation, for the sake of brevity. After equivalent transformations, the anisotropic elastic constants of the fuel and interaction layer can be expressed as
(15)$$\left\{ \begin{array}{l} C_{klmn}^{f}=\lambda^{f}\delta_{kl}\delta_{mn}+\mu^{f}(\delta_{km}\delta_{ln}+\delta_{kn}\delta_{lm})\\ C_{klmn}^{\mathrm{IL}}=\lambda^{\mathrm{IL}}\delta_{kl}\delta_{mn}+\mu^{\mathrm{IL}}(\delta_{km}\delta_{ln}+\delta_{kn}\delta_{lm}) \end{array}\right.$$
where $\lambda^{f}$, $\mu^{f}$, $\lambda^{\mathrm{IL}}$ and $\mu^{\mathrm{IL}}$ denote the Lamé constants of fuel and the interaction layer, respectively.

For the sake of simplification, the mutual negative of the fission pore’s internal pressures are characterized by porosity *f*. Then, substituting Equations (11)–(13), the strain field on the interface Γ of the matrix can be rewritten as
(16)$$\varepsilon^{M}(x,x\in \Gamma )=\int_{\Omega }\left(C^{f}-C^{\mathrm{IL}}):(\varepsilon^{\mathrm{IL}}+f\cdot \varepsilon^{G}+\varepsilon^{H}+\varepsilon^{T}+\varepsilon^{0}\right):\zeta (x,x^{\prime })\;dx^{\prime }$$

The total strain on the interface Γ is deduced by substituting Equations (15) and (16) as (17)$$\begin{array}{cc} \varepsilon^{M}(x,x\in \Gamma )= & \int_{\Omega }\{\left(\lambda^{f}-\lambda^{\mathrm{IL}})\cdot [\delta :(\varepsilon^{\mathrm{IL}}+f\varepsilon^{G}+\varepsilon^{H}+\varepsilon^{T}+\varepsilon^{0})\right]\\ & +2(\mu^{f}-\mu^{\mathrm{IL}})\cdot (\varepsilon^{\mathrm{IL}}+f\cdot \varepsilon^{G}+\varepsilon^{H}+\varepsilon^{T}+\varepsilon^{0})\}:\zeta (x,x^{\prime })\;dx^{\prime } \end{array}$$

Hence, the relation of Equation (15) and the stress field σ on the interface Γ of the matrix can be expressed as
(18)$$\begin{array}{cc} \sigma_{ij}(x,x & \in \Gamma )=\left[\lambda^{M}\delta_{kl}\delta_{ij}+\mu^{M}(\delta_{ki}\delta_{lj}+\delta_{kj}\delta_{il})\right]\int_{\Omega }\{\left(\lambda^{f}-\lambda^{\mathrm{IL}})\delta_{ij}(\varepsilon_{mm}^{\mathrm{IL}}+f\varepsilon_{mm}^{G}+\varepsilon_{mm}^{H}+\varepsilon_{mm}^{T}+\varepsilon_{mm}^{0}\right)\\ & +2(\mu^{f}-\mu^{\mathrm{IL}})(\varepsilon_{ij}^{\mathrm{IL}}+f\varepsilon_{ij}^{G}+\varepsilon_{ij}^{H}+\varepsilon_{ij}^{T}+\varepsilon_{ij}^{0})\}\zeta_{ijkl}(x,x^{\prime })\;dx^{\prime } \end{array}$$

It is noted that the stress of the interior wall equals the pressure exerted on the interior wall for a hollow sphere. The swelling pressure $P_{f}$ of a fuel particle to the matrix should be numerically equal to the radial stress distributed on the interface of the matrix. The irradiation pressure can be expressed as
(19)$$P_{f}=\sigma_{rr}(x,x\in \Gamma )$$

It is observed from Equation (18) that the nonuniformity of swelling pressure comes from the influence function which characterizes the induced relation of eigenstrain on arbitrary position, and anisotropic strains of fuel particles, such as thermal strain, fission gas strain, inhomogeneous strain, etc. Under irradiation, nonuniform swelling pressure can be analyzed as long as the influence function and strains have been determined.

By equivalent transformations of the porous fuel particle model surrounded by an interaction layer, the physical fields of the metal matrix and the swelling pressure exerted on the matrix by fuel particles can be formulated taking into account thermal expansion, the act of gaseous fission fragments, i.e., fission pores, and the act of solid fission fragments, i.e., interaction layer and the mechanical response of three-phase composites. The presented formalism will be analyzed numerically in the next section. Moreover, replacing the fuel particles with nonuniform swelling pressures provides a feasible and simplified way for simulating dispersion fuel elements with porous fuel particles.

## 4. Numerical Examples

### 4.1. Analytical Solution

To analyze the irradiation swelling pressure, the variables and parameters in Equation (18) must be determined. Considering the isotropic linear thermal strain and applied strain, there exist, $\varepsilon^{\mathrm{IL}}=\varepsilon^{\mathrm{IL}}\cdot \delta$, $\varepsilon^{0}=\varepsilon^{0}\cdot \delta$ and $\varepsilon^{T}=\alpha T\cdot \delta$, where α and T denote thermal coefficient of fuel and temperature, respectively. It is noted that the applied strain of the interaction layer equals he thermal strain, that is $\varepsilon^{\mathrm{IL}}=\alpha^{\mathrm{IL}}T$. $\alpha_{\mathrm{IL}}$ denotes the thermal coefficient of the interaction layer. The inhomogeneity strain field has been given [22] for isotropic materials. Let the radius of circular fission pores be expressed as $R_{a}$ the number of fission pores is N with position $x^{\prime }$. Fission gas strain $\varepsilon^{G}$ equals the superposition of each fission pore under internal pressure $P_{g}$ as depicted in Figure 2b, which is presented as [2]
(20)$$P_{g}=\frac{n_{g}RT}{V_{g}}\left(1+\frac{n_{g}a}{V_{g}-n_{g}b}\right)$$
where $n_{g}$ is the total concentration of fission gases. The total volume of fission gases, gas universal constant, temperature, parameters of real gas state equation, density of fuel particles, production of fission gas, burnup, and mole mass of fuel phase, respectively, is denoted by $n_{g}=4\pi R_{f}^{3}D_{f}\beta B_{U}/(3M_{f})$. $V_{g}$, R, T, a, b, $D_{f}$, β, $B_{U}$, $M_{f}$, Let the applied strain and the strain of the interaction layer be isotropic, and Equation (17) can be simplified as
(21)$$\varepsilon^{M}(x\in \Gamma )=\delta :\varepsilon^{\mathrm{IL}}+\alpha T\cdot \delta +\delta :\varepsilon^{0}+\sum_{n=0}^{N}\int_{\Omega }\left\{\lambda^{*}(f\cdot \varepsilon^{G}+\varepsilon^{H})+2\mu^{*}(f\cdot \varepsilon^{G}+\varepsilon^{H})\right\}:\zeta (x,x^{\prime })\;dx^{\prime }$$
where *N* is the number of fission pores. After expansion, Equation (21) can be rewritten as
(22)$$\varepsilon_{ij}^{M}(x\in \Gamma )=\delta_{ij}\varepsilon^{\mathrm{IL}}+\delta_{ij}\alpha T+\delta_{ij}\varepsilon^{0}+\iint_{\Omega }\left[D\zeta_{ij11}(x,x^{\prime })+E\zeta_{ij12}(x,x^{\prime })+F\zeta_{ij21}(x,x^{\prime })+H\zeta_{ij22}(x,x^{\prime })\right]dx^{\prime }$$
where D, E, F, H are parameters which are expressed as
$$D=\lambda^{*}(\varepsilon_{11}^{G}+\varepsilon_{22}^{G}+\varepsilon_{11}^{H}+\varepsilon_{22}^{H})+2\mu^{*}(\varepsilon_{11}^{G}+\varepsilon_{11}^{H})\;E=\lambda^{*}(\varepsilon_{11}^{G}+\varepsilon_{22}^{G}+\varepsilon_{11}^{H}+\varepsilon_{22}^{H})+2\mu^{*}(\varepsilon_{12}^{G}+\varepsilon_{12}^{H})\;F=\lambda^{*}(\varepsilon_{11}^{G}+\varepsilon_{22}^{G}+\varepsilon_{11}^{H}+\varepsilon_{22}^{H})+2\mu^{*}(\varepsilon_{21}^{G}+\varepsilon_{21}^{H})\;H=\lambda^{*}(\varepsilon_{11}^{G}+\varepsilon_{22}^{G}+\varepsilon_{11}^{H}+\varepsilon_{22}^{H})+2\mu^{*}(\varepsilon_{22}^{G}+\varepsilon_{22}^{H})$$

According to Equations (10) and (12), the influence function can be expanded as
(23)$$\begin{array}{ll} \zeta_{ij11}(x,x^{\prime }) & =-\frac{1}{2}(G_{i1,1j}+G_{j1,1i})\\ & =-\frac{1}{2}\left\{\begin{array}{l} \frac{\partial^{2}}{\partial x_{1}\partial x_{j}}\left[\frac{1}{4\pi \mu^{\mathrm{IL}}}\frac{\delta_{i1}}{\left|x-x^{\prime }\right|}-\frac{1}{16\pi \mu^{\mathrm{IL}}(1-\upsilon^{\mathrm{IL}})}\frac{\partial^{2}}{\partial x_{i}\partial x_{1}}\left|x-x^{\prime }\right|\right]\\ \;+\frac{\partial^{2}}{\partial x_{1}\partial x_{i}}\left[\frac{1}{4\pi \mu^{\mathrm{IL}}}\frac{\delta_{j1}}{\left|x-x^{\prime }\right|}-\frac{1}{16\pi \mu^{\mathrm{IL}}(1-\upsilon^{\mathrm{IL}})}\frac{\partial^{2}}{\partial x_{j}\partial x_{1}}\left|x-x^{\prime }\right|\right] \end{array}\right\} \end{array}$$
(24)$$\begin{array}{ll} \zeta_{ij12}(x,x^{\prime }) & =-\frac{1}{2}(G_{i1,2j}+G_{j1,2i})\\ & =-\frac{1}{2}\left\{\begin{array}{l} \frac{\partial^{2}}{\partial x_{2}\partial x_{j}}\left[\frac{1}{4\pi \mu^{\mathrm{IL}}}\frac{\delta_{i1}}{\left|x-x^{\prime }\right|}-\frac{1}{16\pi \mu^{\mathrm{IL}}(1-\upsilon^{\mathrm{IL}})}\frac{\partial^{2}}{\partial x_{i}\partial x_{1}}\left|x-x^{\prime }\right|\right]\\ \;+\frac{\partial^{2}}{\partial x_{2}\partial x_{i}}\left[\frac{1}{4\pi \mu^{\mathrm{IL}}}\frac{\delta_{j1}}{\left|x-x^{\prime }\right|}-\frac{1}{16\pi \mu^{\mathrm{IL}}(1-\upsilon^{\mathrm{IL}})}\frac{\partial^{2}}{\partial x_{j}\partial x_{1}}\left|x-x^{\prime }\right|\right] \end{array}\right\} \end{array}$$
(25)$$\begin{array}{ll} \zeta_{ij21}(x,x^{\prime }) & =-\frac{1}{2}(G_{i2,1j}+G_{j2,1i})\\ & =-\frac{1}{2}\left\{\begin{array}{l} \frac{\partial^{2}}{\partial x_{1}\partial x_{j}}\left[\frac{1}{4\pi \mu^{\mathrm{IL}}}\frac{\delta_{i2}}{\left|x-x^{\prime }\right|}-\frac{1}{16\pi \mu^{\mathrm{IL}}(1-\upsilon^{\mathrm{IL}})}\frac{\partial^{2}}{\partial x_{i}\partial x_{2}}\left|x-x^{\prime }\right|\right]\\ \;+\frac{\partial^{2}}{\partial x_{1}\partial x_{i}}\left[\frac{1}{4\pi \mu^{\mathrm{IL}}}\frac{\delta_{j2}}{\left|x-x^{\prime }\right|}-\frac{1}{16\pi \mu^{\mathrm{IL}}(1-\upsilon^{\mathrm{IL}})}\frac{\partial^{2}}{\partial x_{j}\partial x_{2}}\left|x-x^{\prime }\right|\right] \end{array}\right\} \end{array}$$
(26)$$\begin{array}{ll} \zeta_{ij22}(x,x^{\prime }) & =-\frac{1}{2}(G_{i2,2j}+G_{j2,2i})\\ & =-\frac{1}{2}\left\{\begin{array}{l} \frac{\partial^{2}}{\partial x_{2}\partial x_{j}}\left[\frac{1}{4\pi \mu^{\mathrm{IL}}}\frac{\delta_{i2}}{\left|x-x^{\prime }\right|}-\frac{1}{16\pi \mu^{\mathrm{IL}}(1-\upsilon^{\mathrm{IL}})}\frac{\partial^{2}}{\partial x_{i}\partial x_{2}}\left|x-x^{\prime }\right|\right]\\ \;+\frac{\partial^{2}}{\partial x_{2}\partial x_{i}}\left[\frac{1}{4\pi \mu^{\mathrm{IL}}}\frac{\delta_{j2}}{\left|x-x^{\prime }\right|}-\frac{1}{16\pi \mu^{\mathrm{IL}}(1-\upsilon^{\mathrm{IL}})}\frac{\partial^{2}}{\partial x_{j}\partial x_{2}}\left|x-x^{\prime }\right|\right] \end{array}\right\} \end{array}$$

Let the parameters be given for convenience as $I=\frac{1}{4\pi \mu^{\mathrm{IL}}}$, $K=I\cdot \frac{1}{4(1-\upsilon^{\mathrm{IL}})}$ and $X={\left|x-x^{\prime }\right|}^{2}$, the strain on interaction Γ of matrix can be expressed as,
(27)$$\varepsilon_{11}^{M}(x,x\in \Gamma )=\varepsilon^{\mathrm{IL}}+\varepsilon^{0}+\alpha_{f}T-\int_{\theta_{1}}^{\theta_{2}}\int_{r_{1}}^{r_{2}}\left[\begin{array}{l} D\cdot \frac{\partial^{2}}{\partial x_{1}^{2}}(I\cdot \frac{1}{X}-K\cdot \frac{\partial^{2}}{\partial x_{1}^{2}}X)+E\cdot \frac{\partial^{2}}{\partial x_{2}\partial x_{1}}(I\cdot \frac{1}{X}-K\cdot \frac{\partial^{2}}{\partial x_{1}^{2}}X)\\ \;+F\cdot \frac{\partial^{2}}{\partial x_{1}^{2}}(-K\cdot \frac{\partial^{2}}{\partial x_{1}\partial x_{2}}X)+H\cdot \frac{\partial^{2}}{\partial x_{2}\partial x_{1}}(-K\cdot \frac{\partial^{2}}{\partial x_{1}\partial x_{2}}X) \end{array}\right]dx_{1}^{\prime }dx_{2}^{\prime }$$
(28)$$\varepsilon_{22}^{M}(x,x\in \Gamma )=\varepsilon^{\mathrm{IL}}+\varepsilon^{0}+\alpha_{f}T-\int_{\theta_{1}}^{\theta_{2}}\int_{r_{1}}^{r_{2}}\left[\begin{array}{l} D\cdot \frac{\partial^{2}}{\partial x_{1}\partial x_{2}}(K\cdot \frac{\partial^{2}}{\partial x_{2}\partial x_{1}}X)+E\cdot \frac{\partial^{2}}{\partial x_{2}^{2}}(K\cdot \frac{\partial^{2}}{\partial x_{2}\partial x_{1}}X)\\ \;+F\cdot \frac{\partial^{2}}{\partial x_{2}\partial x_{1}}(-I\cdot \frac{1}{X}+K\cdot \frac{\partial^{2}}{\partial x_{2}^{2}}X)+H\cdot \frac{\partial^{2}}{\partial x_{2}^{2}}(-I\cdot \frac{1}{X}+K\cdot \frac{\partial^{2}}{\partial x_{2}^{2}}X) \end{array}\right]dx_{1}^{\prime }dx_{2}^{\prime }$$
(29)$$\begin{array}{ll} \varepsilon_{12}^{M}(x,x\in \Gamma ) & =\varepsilon_{21}^{M}(x,x\in \Gamma )\\ & =-\frac{1}{2}\int_{\theta_{1}}^{\theta_{2}}\int_{r_{1}}^{r_{2}}\left\{\begin{array}{l} \frac{\partial^{2}}{\partial x_{1}\partial x_{2}}\left[-(D+H)\cdot I\cdot \frac{1}{X}+K\left(D\frac{\partial^{2}}{\partial x_{1}^{2}}X+H\frac{\partial^{2}}{\partial x_{2}^{2}}X+(F+E)\frac{\partial^{2}}{\partial x_{1}\partial x_{2}}X\right)\right]\\ \;+\frac{\partial^{2}}{\partial x_{2}^{2}}\left[-E\cdot I\cdot \frac{1}{X}+K(E\frac{\partial^{2}}{\partial x_{1}^{2}}X+H\frac{\partial^{2}}{\partial x_{1}\partial x_{2}}X)\right]\\ \;+\frac{\partial^{2}}{\partial x_{1}^{2}}\left[-F\cdot I\cdot \frac{1}{X}+K(D\cdot \frac{\partial^{2}}{\partial x_{2}\partial x_{1}}X+F\frac{\partial^{2}}{\partial x_{2}^{2}}X)\right] \end{array}\right\}dx_{1}^{\prime }dx_{2}^{\prime } \end{array}$$

The swelling pressures $P_{f}$ of fuel particles to matrix should be numerically equal to the radial stresses distributed on the interface of the matrix. As a consequence, nonuniform irradiation swelling pressure $P_{f}$ can be calculated by substituting Equation (26) into (18). Under irradiation, the swelling pressure exerted on the matrix in fuel meat is a function of temperature T, burnup $B_{U}$, geometric coordinates $x_{i}$, fuel particle radius $R_{f}$, thickness of interaction layer $d_{\mathrm{IL}}$, the material constants of fuel $\mu^{f}$, $\lambda^{f}$, interaction layer $\mu^{\mathrm{IL}}$, $\lambda^{\mathrm{IL}}$ and matrix $\mu^{M}$, $\lambda^{M}$,
(30)$$P_{f}=P_{f}(T,B_{U},x_{i},R_{f},d_{\mathrm{IL}},\mu^{f},\lambda^{f},\mu^{\mathrm{IL}},\lambda^{\mathrm{IL}},\mu^{M},\lambda^{M})$$

With the assumption of isotropic linear thermal strain, the nonuniform swelling pressures exerted on the matrix have been induced by inhomogeneous distribution of fission pores. Once the location of every pore is determined, swelling pressure can be analyzed.

### 4.2. Numerical Results and Discussions

Three examples are investigated analytically and simulated to investigate the factors of influence and to validate the accuracy of the proposed formalism. The three simplified models have one pore, two symmetric pores and four symmetric pores, respectively. Finite element analysis was carried out for the three models by using the program ABAQUS/CAE. Symmetric boundary conditions were applied on the boundary of the matrix in finite simulations. Meanwhile, as depicted in Figure 2, the boundary of the matrix was subjected to an environmental pressure of 15 MPa. Accounting for the conditions of 300 °C and 0.2 FIMA, the internal pressures of fission pores were obtained from Equation (20) [2]. The material properties of stainless steel [30] and UO_2_ ceramics [31] at 300 °C were applied for the matrix and fuel particles, respectively. Compared with thermal expansion, the observation and analysis implied small creep strains in the hypothetical steady-state irradiation creep regime of ceramics [32]. The material properties of stainless steel were tested after irradiation so that the creep effect was included in material plasticity. For calculation and simulation, the thickness of the interaction layer was set as 9.1 μm [33,34]. The distributions of irradiation swelling pressures exerted on the matrix interface Γ could be calculated according to the presented formalism and simulated by the finite element method. Comparisons of analytical and simulated irradiation swelling pressures exerted on the matrix interface Γ, i.e., along the path ABCDA, are depicted in Figure 4. It can be observed that there were nonuniform swelling pressures exerted on the matrix. The analytical swelling pressures matched well with the simulated values. Accordingly, the formalism presented was validated as being accurate and could be adopted to analyze more problems under different conditions. The swelling pressures of fuel particles to the matrix possessed nonuniformity under irradiation.

As shown in Figure 4a, the irradiation swelling pressure $P_{f}$ is uniformly exerted on the matrix when there is a central fission pore. As for when there were two symmetric fission pores, the nonuniform distribution of the swelling pressure is presented in Figure 4b. There were two peak values that emerged at the location closed to two pores. Analogously, Figure 4c demonstrates that four peak values emerged at the location closed to four pores.

Consequently, there were substantial effects of the numbers and locations of fission pores on nonuniform irradiation swelling pressure. With increasing pore numbers, the nonuniformity rises. The swelling pressure appears to surge at the location where the fission pores are closed to the matrix. The peak value is associated with distances between fission pores and the interface. With same distances, the number of peak values increases with numbers of fission pores, while the magnitudes of all peak values remain constant. The nonuniform swelling pressures can be analyzed if the locations of fission pores are determined.

Figure 5 depicts the effects of burnup and temperature on nonuniform swelling pressures when there are two symmetric pores. With increase in burnup, the average swelling pressure remains invariant while peak values ascend. With respect to temperature, the average swelling pressures increased with increasing irradiation temperature. Nonetheless, under the same burnup, the peak values hardly changed with variation in temperature.

As commonly known, the interior pressures inside fission pores are mainly influenced by burnup. Hence, the peak values alter greatly with rising burnup. On the other hand, due to the assumption of isotropic linear thermal expansion, average swelling pressure was affected by temperature. Notwithstanding, it was noted that the maximum of irradiation swelling pressure under high temperature is larger than under high burnup. The nonuniformity of swelling pressures appears to be aggravated under high burnup, rather than high temperature. Accordingly, high temperature tends to be more deleterious for the integrity of the matrix under irradiation. For safety of fuel elements, it is more dangerous to have high temperature than high burnup.

It is noted that failure in dispersion fuel meat originates from the interaction layer and propagates to the matrix. It is the initial damages and failures that distribute at the location with maximum stress. As a consequence, the nonuniformity analysis of swelling pressure exerted on the matrix is considerable. Through calculating distributions of maximum principal stresses by means of the presented formalism, it is convenient to investigate the failure of dispersion fuel meat further.

### 4.3. The Numerical Analysis of a Porous Fuel Particle

The fission pores in fuel particles are generated unevenly under irradiation. The swelling pressures $P_{f}$ of the porous fuel particle to matrix can be calculated according to the upper analytical process. Meanwhile, the physical fields in a metal matrix can also be constructed and analyzed numerically. Similarly, with the presented formalism, the strain field of the metal matrix is directly expressed as
(31)$$\varepsilon^{M}(x)=\int_{\Omega }(C^{M}-C^{\mathrm{IL}}):\varepsilon^{**}:\;\zeta (x,x^{\prime })\;dx^{\prime }$$

Accounting for the dangerous conditions of 500 °C and 0.3 FIMA, the distributions of pores in a porous fuel particle submit to Weibull distribution, as shown in Figure 6a. The fission porosity is 15 percent. The internal pressure of the fission pore is obtained from Equation (20) [2] in terms of the material properties of stainless steel [30] and UO_2_ ceramics [31] at 500 °C. The swelling pressures $P_{f}$ of the fuel particle to the matrix should be numerically equal to the radial stress on the matrix interface Γ. The radial stress state of the matrix can be calculated according to Equation (31). Removing the fuel particle, the contour diagram of radial stress in the matrix is demonstrated as Figure 6b. It is observed that there is a nonuniform stress state on the matrix interface Γ. Consequently, there is great effect on the stress state in dispersion fuel meat due to porous fuel particles under irradiation.

For the porous structure of Figure 6a, the swelling pressure exerted on the matrix can be calculated according to Equation (18), as depicted in Figure 7. Therefore, on the basis of the proposed formalism, the pressures exerted on the matrix by a fuel particle swelling can be numerically calculated for dispersion fuel meat under different irradiation conditions. Furthermore, the irradiation-induced swelling pressures can be adopted to investigate strength and failure of the matrix. It is noted that the simulated investigation of dispersion fuel elements with porous fuel particles is difficult to converge and compute. When replacing the act of porous fuel particles with nonuniform swelling pressure exerted on the matrix, the macro fuel element model possesses better convergence and computation.

Explicit analysis of nonuniform irradiation-induced pressure exerted on the dispersion fuel matrix by a fuel particle swelling is expected to analyze the failure problems of dispersion fuel meat and provide a calculable way to investigate fuel elements. The formalism presented in this study, which is applied for ceramic fuel, can be further developed for other inelastic fuels. Accordingly, the presented analytical method about irradiation-induced swelling pressure exerted on dispersion fuel matrix lays the foundation for analyzing the failure of dispersion fuel meat, reducing computational effort and possessing better convergence in macroscopic simulation of a dispersion fuel element with porous fuel particles.

## 5. Conclusions

In this study exploring the effects that lead to the failure of dispersion fuel meat, nonuniform irradiation-induced pressure exerted on the metal matrix by ceramic fuel swelling was formulated on the basis of the equivalent inclusion method and calculated numerically with the aid of Green’s function. Several examples have been given to demonstrate the application of the presented formalism and to investigate the influencing factors of nonuniformity. The main conclusions are as follows:

The nonuniform irradiation swelling pressure $P_{f}$ exerted on the matrix can be analytically formulated and calculated, based on the equivalent inclusion method. By comparing the analytical results and the simulated results, the proposed formalism is validated to be relatively precise.

The numbers and distributions of fission pores inside a fuel particle are the primary factors affecting the uniformity of swelling pressures. It is identified that swelling pressures surge at the location where the fission pores are closed to the matrix. The stress states in the metal matrix can be analyzed once the distributions of fission pores are determined under irradiation.

The swelling pressure increases with rising temperature and burnup. The nonuniformity of swelling pressure appears to be aggravated under high burnup more than high temperature. The maximum increases more under high temperature than under high burnup.

Based on the proposed formalism, the swelling pressures exerted on the matrix interfaces and the stress states in the metal matrix can be numerically analyzed for porous fuel particles under different irradiation conditions. The nonuniformity analysis of swelling pressures can be employed to investigate the failure, strength and integrity of dispersion fuel meat.

## Figures and Tables

**Figure 1 materials-15-03231-f001:**
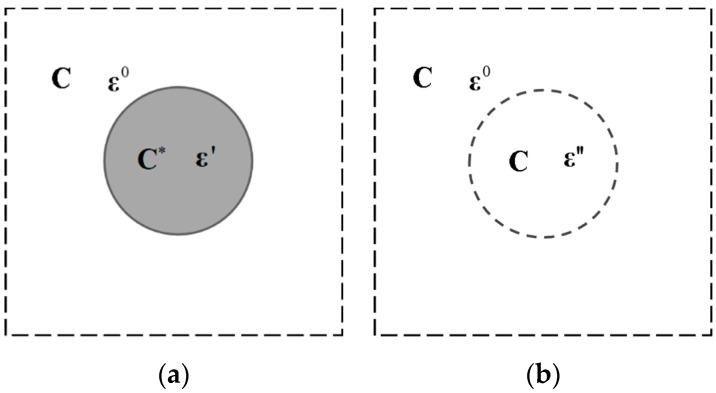
Equivalent transformation of (**a**) an inhomogeneous inclusion to (**b**) a homogeneous inclusion subjected to eigenstrain
$\varepsilon^{\prime \prime }$.

**Figure 2 materials-15-03231-f002:**
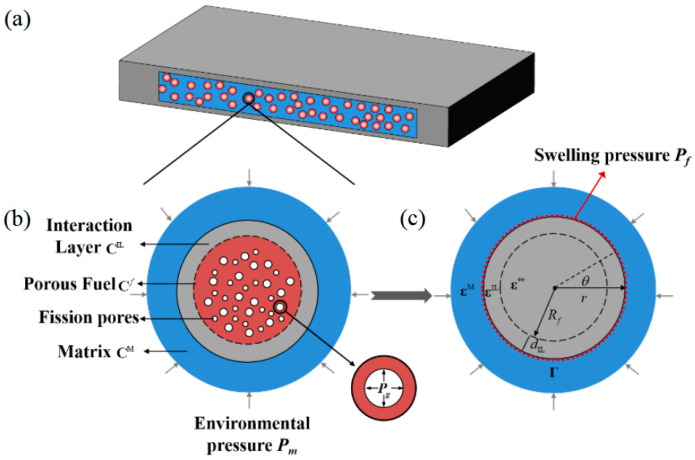
Equivalent transformation of single particle representative volume element. (**a**) Dispersion fuel element; (**b**) Single particle representative volume element with three-phase components under irradiation; (**c**) The transformed mechanical model in which swelling pressure is exerted on the interface Γ.

**Figure 3 materials-15-03231-f003:**
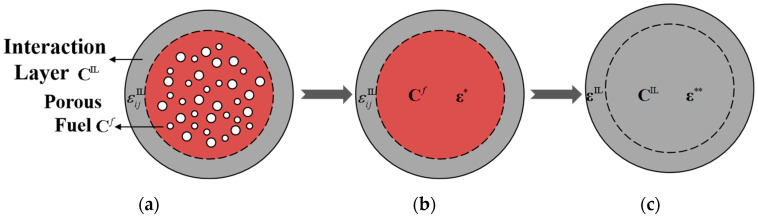
Double transformations according to the equivalent inclusion method. (**a**) A porous fuel particle surrounded by interaction layer; (**b**) An homogeneous equivalent inclusion surrounded by interaction layer after the first transformation; (**c**) An homogeneous equivalent inclusion after the second transformation.

**Figure 4 materials-15-03231-f004:**
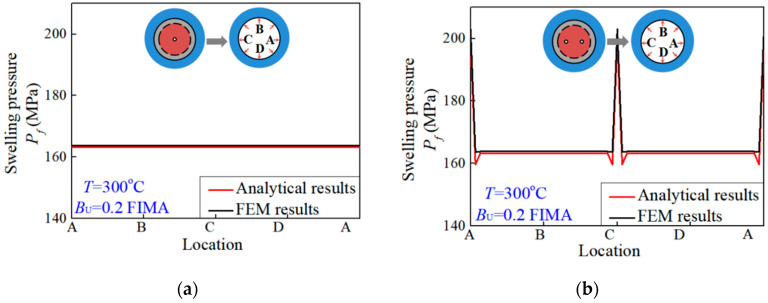
The distributions of irradiation swelling pressures exerted on the matrix interface Γ when there exists (**a**) one central fission pore, (**b**) two symmetric fission pores and (**c**) four symmetric fission pores.

**Figure 5 materials-15-03231-f005:**
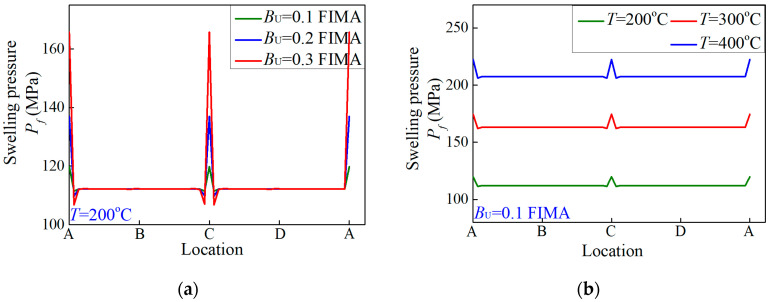
The distributions of irradiation swelling pressures exerted on the matrix interface Γ when there were two symmetric fission pores under (**a**) different burnup and (**b**) different temperatures.

**Figure 6 materials-15-03231-f006:**
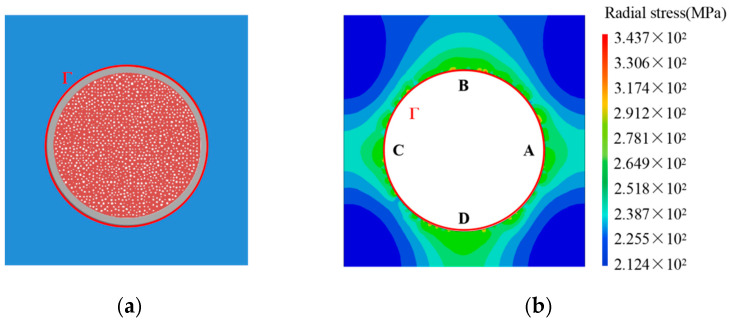
Analytical results for the irradiation swelling pressure of a porous fuel particle at dangerous conditions of 500 °C and 0.3FIMA. (**a**) Structure of a porous fuel particle; (**b**) Contour diagram of radial stress in the matrix.

**Figure 7 materials-15-03231-f007:**
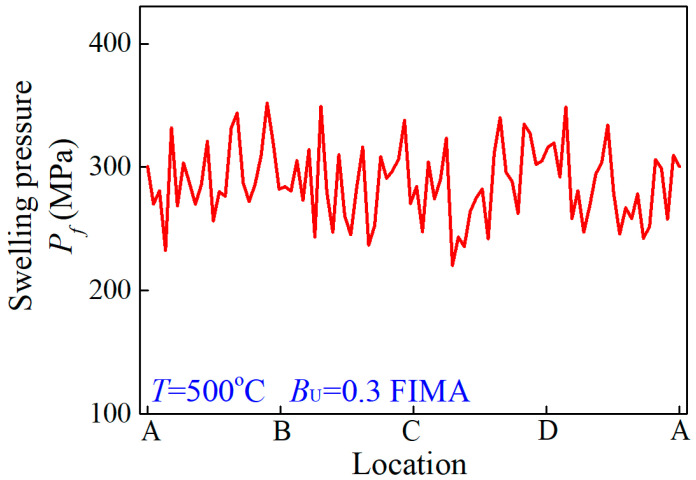
The swelling pressure on matrix interface Γ at dangerous conditions of 500 °C and 0.3 FIMA.

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
