# Peer review of "Explicit Analysis of Nonuniform Irradiation Swelling Pressure Exerting on Dispersion Fuel Matrix Based on the Equivalent Inclusion Method"

_materials, 2022, doi:10.3390/ma15093231_

Round 1
Reviewer 1 Report
The manuscript represents a very interesting study on the nonuniform irradiation swelling pressure under different irradiation conditions by applying the so-called equivalent inclusion method wich allows the transofrmation of inhomogeneous inclusion to homogeneous inclusion. The authors studied in particular the nonuniform irradiation swelling pressure where very impressive results are obtained. The results are also compared with those fromm the finite element method. The model is clearly presented. The results and conclusions of the paper reads sound to me. And the mansucript is well written. I support its prompt publication.
A few minor comments for the authors to consdier:
" twice equivalent transformations" I personally recommend to call it something like doubly
equivalent transformations
", the size effect via the concept of strain gradient 60
elasticity[20], etc" This is confusing statement.
"As a consequence," of what? It reads strange to have a paragraph starting with that. And the fact that the method can be effectively employed is not a "consequence" of those mentioend in above paragraph but maybe better rephrased as something like an advantage.
"Creep was not considered in consti- 79
tutive relationship. But the material properties in simulation was creep-dependent. " It's so strange and confusing to read those two statements in the Introduction? I guess what the authors wanted to say is that creep was not consdiered before?(if so, a reference needed) but it can be included in the present study??
"some conclu- 91
sions are addressed" -->maybe better to say, the conclusions are drawn unless the author want to address/comment on earlier conclusions.
"in order to explore the failure of dispersion fuel meat," may be more specifically stated it as "to explore the effects that lead to the failure of ..."
Reviewer 2 Report
Dear Authors,
the paper proposes a rather interesting mathematical model to predict the nonuniform irradiation swelling pressure exerting on a fuel matrix under irradiation.
The manuscript is well written and the results look consistent.
Some parts of the paper need some additional clarifications, according to the following comments:
Introduction
1) you mention in the description of the previous researches on the subject that the fuel inhomogeneities can be already properly modelled. From the manuscript, it is not clarified which are the limitations of the current approaches and how your study may solve such limitations and/or improve the current modeling capabilities. Please, add some sentences on that.
2) you mention that creep is not considered in your approach. Please, explain why: its effect is minor? or it cannot be modelled?
Paragraph 4.2
a verification of the proposed approach is performed against 'simulations'. Which tool has been used for the simulations, which are considered in the context of your manuscript as reference for the approach you propose?
Going to the results shown in Figure 4. Since mathematical models are employed in the simulation tools, the very good agreement between the analytical solution and the simulations means that the simulation tool itself is already capable to properly predict the fuel behavior. then again the above question: what is the added value of your approach?
Is your approach valid for a burnup range larger than 0.3 FIMA?
Conclusions
Lines 434-437. The sentences are in line with the results presented in the paper. Is any validation activity (namely against experiments) envisaged in the future?
Best Regards
